# Stratospheric Aerosols: Establishing a Novel Optical Thickness Benchmark for Effective Climate Change Mitigation

## Abstract

Global Warming has been a problem at the heart of Earth's environmental issues for nearly 5 decades, with the potential to affect a significant portion of the global population and cause catastrophic irreversible damage to the planet's future. Changes in Earth's climate due to the rise in global temperatures will have an enormous impact on communities around the world, along with a drastic displacement of humans and an extreme loss in natural biodiversity. Current methods of combating this issue have proven to be ineffective, requiring a more comprehensive and innovative approach. This project aims to propose a potential solution to mitigate the effects of global warming and limit temperatures to sustainable levels through the use of stratospheric aerosols. Through a process of data collection, experimentation, and modeling, I was able to correlate the presence of aerosols in the stratosphere to a consequent drop in temperatures and utilize regression prediction to forecast a 16 percent drop in global temperatures after examining the effects of volcanic ash in the stratosphere. I was also able to compare monthly aerosol concentration levels to declines in the growth of temperatures and conclude that by keeping aerosol optical thickness over 0.185, we can stabilize global temperatures and achieve climate change goals set to protect the Earth. By implementing the changes to Earth's atmosphere, we can reflect heat from the Sun and create a cooling effect for the planet, potentially stopping climate change and saving billions of people.

## 1 Introduction

Conventional approaches to combat climate change show little change in annual temperature growth trends. Stratospheric aerosols offer potential for limiting global warming through solar dimming, yet their impact remains understudied. This paper proposes a holistic approach to assess stratospheric aerosols' efficacy globally. Historical evidence, such as the Laki eruption in 1783, demonstrates the potential impact in reducing temperatures. Recent major volcanic eruptions led to significant aerosol deposition, shown by a global decrease in optical depth. This research explores aerosol concentrations' impact on global temperatures and proposes guidelines for utilizing stratospheric aerosols to reflect excess sunlight and limit global warming.

## 2 Related Works

Aerospace regulations have made it challenging to experiment with aerosols in the atmosphere. Some groups have investigated the idea of stratospheric aerosol injection (SAI) to combat climate change, which involves injecting artificial particles into the atmosphere to block sunlight. The paper by Schmeisser et al. (2017), discusses suitable aerosol materials, while Ramanathan (2001) explores the potential impacts of SAI on the climate using numerical models. Research on the feasibility of SAI has been limited, with much focus on cost and resources required, as shown in the paper by Moriyama et al. (2016). My paper proposes a novel approach to assess the efficacy of stratospheric aerosols using satellite data to analyze their global impacts.

## 3 VOLCANIC AEROSOL EMISSIONS ON TEMPERATURES

NASA's MERRA-2 Dataset and the Food and Agriculture Organization offer global data on the amount of solar light lost due to stratospheric aerosols. Aerosols from events such as the El Chichón eruption in 1982 can be seen in data many months after the initial eruption. Not all powerful volcanic eruptions affect the measurement, however. This typically occurs when the maximum plume height from an eruption fails to exceed the 10-mile stratosphere layer, as in the case of the Chaitén eruption in 2008. Major volcanic eruptions cause global temperature drops due to solar dimming, in which stratospheric aerosols reflecting sunlight back into space. See Figure 2.

Using regression, we predict a 1.674 C temperature increase by 2030. However, analyzing temperature data following major eruptions shows a flatter trend. KNN imputation fills missing values in the data set, allowing us to predict a temperature of 1.411 C in 2030 with volcanic aerosols, a 16 percent drop from general warming. This aligns with the Paris Agreement's temperature limits for sustained world growth, as it is within the 1.5 C limit. See Figure 3.

## 4 AEROSOL OPTICAL DEPTH LIMIT

The GACP releases monthly concentration levels of aerosols in the stratosphere over Earth's oceans from 1983 to 1997, providing N=185 samples to examine. Averaging the 1x1 global grid boxes of data per month gives the mean level of global aerosol optical depth. Comparing each of these values with monthly changes in global temperature shows a clear relationship between higher levels in global average aerosol concentration and drops in temperatures. By quantifying this relationship, a value of 0.185 is found to represent the average thickness of the aerosol concentration at which there is typically a drop in global temperatures. Analyzing samples over the 0.185 benchmark results in an average drop in global temperatures of 0.279 C. This suggests that when the thickness of the aerosol layer in the atmosphere is above 0.185, there is a significant decrease in the warming of the planet.

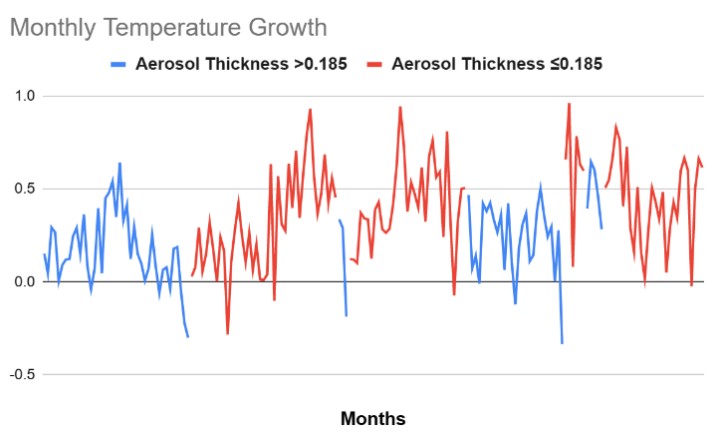

Figure 1: Graph of monthly aerosol levels and warming

## 5 CONCLUSIONS

The study shows the potential of using aerosols to combat climate change by analyzing the impact of volcanic aerosols on global temperatures. Regression techniques were used to forecast the magnitude of global warming with and without the presence of volcanic aerosols, resulting in a 16 percent drop in predicted temperature in 2030. Through statistical analysis, it was concluded that keeping aerosol thickness over the proposed 0.185 benchmark would decrease the global temperature. The study underscores the critical importance of stratospheric aerosols in mitigating the effects of climate change and safeguarding the future of the planet.

URM STATEMENT

The authors acknowledge that at least one key author of this work meets the URM criteria of ICLR 2023 Tiny Papers Track.

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

## A   APPENDIX

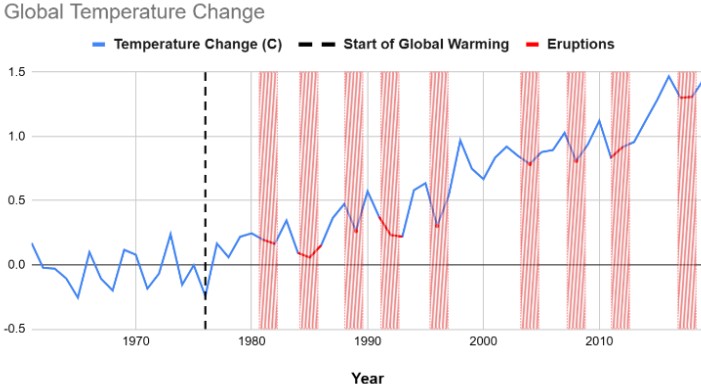

Figure 2: Graph of yearly warming highlighting volcanic activity

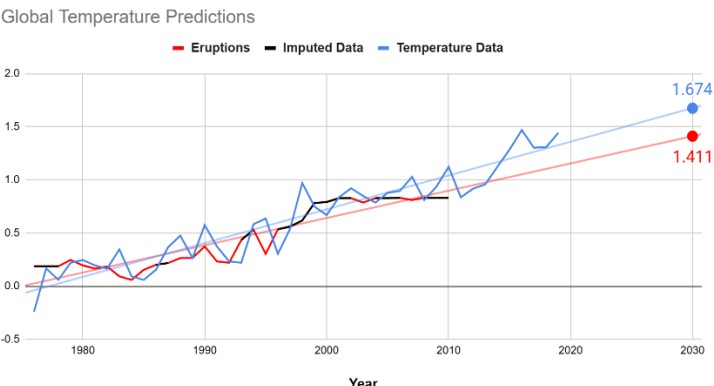

Figure 3: Graph of predicted warming levels in 2030

