# OpenReview forum: "Stratospheric Aerosols: Establishing a Novel Optical Thickness Benchmark for Effective Climate Change Mitigation"
_ICLR.cc/2023/TinyPapers — Submitted to Tiny Papers @ ICLR 2023_

### Official Review · Reviewer_n9ks · 2023-03-30

**Confidence:** 4

**Summary Of Contributions:**

This paper tries to identify the effects of global warming and limit temperatures to sustainable levels using stratospheric aerosols. Focusing on data analysis and experiments, the authors correlate the presence of aerosols in the stratosphere to a consequent drop in temperatures and utilize regression models to forecast a 16% drop in global temperatures after examining the effects of volcanic ash in the stratosphere.

**Rating:**

Needs Clarification (NC): a submission which does not meet the reviewing criteria and needs clarification for its described problem or solution

**Strengths And Weaknesses:**

The authors compare the monthly aerosol concentration levels to declines in the growth of temperatures and conclude that by keeping aerosol optical thickness over 0.185, one can stabilize global temperatures and achieve climate change goals set to protect the Earth. However, in terms of reproducibility, no details are provided on the regression models, and data preprocessing, while a comparison with other methods is missing. The significance of the contribution lacks motivation and experimental evidence. At the same time, no theoretical insights/methods are presented - the analysis is limited to a few experiments.

**Suggested Changes:**

Replace statements starting with “I was able” with “We…” and “My paper..” with “Our study..” for improving the formal academic style of the paper.
Would be helpful to Include a section that clearly describes the methods examined in the paper and the experimental setting. More detailed comparisons with relevant works and the difference/advantages of the proposed method could also support the experimental results.

---

### Official Review · Reviewer_EZQL · 2023-03-30

**Confidence:** 3

**Summary Of Contributions:**

This paper showcases the positive benefit of thick stratospheric aerosols on global temperatures. This paper claims to use regression (without going into details)  to forecast the potential impact of these aerosols on global temperatures.

**Rating:**

Needs Clarification (NC): a submission which does not meet the reviewing criteria and needs clarification for its described problem or solution

**Strengths And Weaknesses:**

Strength:
1. This paper tackled an important problem.
2.  This paper proposed benchmark of optical thickness, without going into details.



Weakness:
1. No mention of how this paper adds value to existing literature. Perhaps citing more paper will help.
2. Regression is used very loosely without going into further details.

**Suggested Changes:**

Suggested Changes:
1. Make this paper reproducible, by talking about dataset and talking about your contribution.
2. Connect it to the existing literation by citing papers.

---

### Official Review · Reviewer_6afF · 2023-04-03

**Confidence:** 5

**Summary Of Contributions:**

This paper investigates the potential of stratospheric aerosols for mitigating climate change by establishing an optical thickness value (a measure of atmospheric aerosol content, in some way) that is necessary to observe a drop in global temperatures. The study uses historical data, statistical analysis, and regression prediction to claim that maintaining an aerosol optical thickness of 0.185 could stabilize global temperatures.

**Rating:**

Great Start (GS): a submission which meets some of the reviewing criteria but has room for improvement

**Strengths And Weaknesses:**

Strengths:

1. The paper tries to quantify the impact of aerosols on global temperatures - a problem of practical interest and of urgent global concern.
2. The use of historical data and information of volcanic eruptions provides appropriate context.

Weaknesses:

1. While the paper tries to establish the extent of aerosol needed in atmosphere (using aerosol optical depth as the metric) to stabilize global temperatures and meet the Paris Agreement temperature limits, it's not clear from the title or the abstract and introduction. I think the text in abstract such as "This project aims to propose a potential solution to mitigate the effects of global warming and limit temperatures .." is misleading.

2. The paper lacks a comprehensive discussion on the potential side effects, risks, and ethical considerations of stratospheric aerosol injection - I think it is essential to include this while proposing a controversial solution to a problem that can have far-reaching effects. (see suggestions if space was a concern).

3. The literature review can be more comprehensive and a lot of content in the paper needs appropriate citations, especially while using factual information (e.g., the Paris agreement temperature limits info should have a citation).

4. While some information is provided about the analysis and the data used, I don't think it is enough to reproduce the results. For example, there is no code provided and no information is provided on where the global temperature data is sourced from.

5. The paper contains some grammatical and formatting issues that reduce its clarity and overall presentation - see suggestions for specific examples.


**Suggested Changes:**

1. I think every paper, no matter how small, should be self-contained. The terms 'Optical Thickness' and 'Optical Depth Limit' were used in the title and throughout the paper but never sufficiently defined. While I can infer with a bit of Googling that these are in some way indicators of atmospheric aerosol levels I think they should be defined in the paper.

2. The authors can make additional space needed for suggestions by cutting down some of the existing text. For example, the abstract can be a lot shorter - some of the elements in the abstract should instead be in the introduction. Let abstract provide a gist of the work.

3. On a personal level, I think the use of words 'data collection' and 'experimentation' in the abstract line "Through a process of data collection, experimentation, and modeling, I was able to correlate the presence of aerosols in the stratosphere to a consequent drop in temper ..." is a bit misleading. I assumed that you collected data yourself and did physical experiments. I think the work is on modeling and analysis using available open source data sources and that should be clarified in the paper.

4. The authors switch between first 'I', 'Myself', and 'we' in the text often. I believe even if it is a single author, the norm is to use
'we'. Either way, I think it should be consistent through out the text.

5. I think the authors should include a more thorough discussion of related literature. Projections on temperature rise and associated metrics are heavily studied, including all factors, and I think the current work doesn't do a good job at summarizing related work. A linear regression model is a great first step, but I think any work at a level of proposing this as an alternative solution should build upon current state of the art. For example [1] the inherent uncertainty involved with any projections using aerosols.

6. I think the title should be more clear - as far as I know, the word benchmark is usually used in a completely different context and might be a bit misleading.

7. As I mentioned earlier, in a work that makes claims based on empirical observations, the authors should provide the source code and the data to reproduce the results.

8. I think Figure 1 should have the scale on x-axis. It is not entirely clear if the occasional peaks are due to seasonal increases but it would have been clear if perhaps months were clearly marked on the axis.

[1] Watson-Parris, Duncan, and Christopher J. Smith. "Large uncertainty in future warming due to aerosol forcing." Nature Climate Change (2022): 1-3.

---

### Meta-Review · Area_Chair_iob8 · 2023-04-05

**Recommendation:** Invite to revise
**Confidence:** 5

**Metareview:**

This paper showcases the positive benefit of thick stratospheric aerosols using historical data, statistical analysis, and regression prediction, and asserts that an optical thickness of 0.185 can stabilize temperatures. The authors analyze the correlation between aerosols and temperature drops, and forecast a 16% decrease in global temperatures after examining the impact of volcanic ash in the stratosphere.

**Summary:**

This paper explores the potential of stratospheric aerosols in mitigating climate change by identifying an aerosol optical thickness value needed to reduce global temperatures.

**Reason For Not Giving A Higher Recommendation:**

I agree with the reviewers that the paper lacks clarity in its title, abstract, and introduction, and needs a more comprehensive discussion of side effects, risks, and ethical considerations. It has gaps in citations and literature review, and doesn't provide sufficient information to reproduce results. The presentation is affected by grammatical and formatting issues. Improvements in these areas would strengthen the paper.

**Reason For Not Giving A Lower Recommendation:**

N/A

---

### Decision · Program_Chairs · 2023-04-10

No revision received; not invited to archive